# Rectal Sensory and Compliance Testing: A Method Comparison Study between High-Resolution Anorectal Manometry and Barostat Investigations

**DOI:** 10.3390/diagnostics14040351

**Published:** 2024-02-06

**Authors:** Lucian Marinica Grando, Jonas Halfvarson, Michiel van Nieuwenhoven

**Affiliations:** 1Department Gastroenterology, Faculty of Medicine and Health, Örebro University, SE 7082 Örebro, Sweden; jonas.halfvarson@regionorebrolan.se (J.H.); michiel.van-nieuwenhoven@regionorebrolan.se (M.v.N.); 2University Health Care Research Center, Faculty of Medicine and Health, Örebro University, SE 70182 Örebro, Sweden

**Keywords:** rectal sensory function, rectal compliance, high-resolution anorectal manometry, rectal barostat

## Abstract

Abnormal visceral perception and motor function are often observed in patients with fecal incontinence, evacuation disorders and irritable bowel syndrome. The international anorectal physiology working group has proposed a standardization for anorectal function assessment, where rectal sensitivity testing is performed using an elastic balloon attached to a high-resolution anorectal manometry (HRAM) catheter. Rectal compliance, another component of rectal function evaluation, is a pressure–volume relationship that refers to the rectum’s ability to stretch and expand as it receives and holds fecal matter. There are no data available regarding the possibility of compliance testing using HRAM, although this is theoretically possible by correcting for the elastic balloon’s intrinsic properties. The gold standard for measurement of visceral sensitivity and compliance is the rectal barostat, according to the procedure described by the European COST action GENIEUR group. Data on the agreement between the two different procedures are scarce. Hence, we performed a comparative study of the HRAM and barostat investigations in 26 healthy individuals. We hypothesized that by inflating the balloon before the examination, rectal compliance can be measured with HRAM investigations, and we examined correlations and levels of agreement between the methods. Our results demonstrate that assessing rectal compliance with HRAM is technically possible; however, a strong correlation with the rectal barostat was only observed at the maximum tolerable volume (Spearman’s rho = 0.7, *p* = 0.02). We only found moderate correlations (Spearman’s rho = 0.562, *p* = 0.019) for compliance according to the barostat methodology and for rectal sensibility testing (Spearman’s rho = 0.57, *p* = 0.03 for maximum tolerable volume). Bland–Altman plots showed poor levels of agreement between the methods. We conclude that HRAM and the rectal barostat cannot be used interchangeably for compliance or sensitivity assessments. We suggest the development of a non-elastic balloon with a fixed size and shape to assess rectal sensory function and compliance in HRAM testing.

## 1. Introduction

Rectal sensory and motor functions are important for normal bowel physiology. Abnormal rectal sensitivity or biomechanical function (usually defined as rectal compliance) is frequently observed in fecal incontinence, evacuation disorders and irritable bowel syndrome, justifying their evaluation in clinical practice [1,2,3,4]. 

There are different techniques to measure rectal sensitivity and compliance, but all are based on the insertion and subsequent inflation of a balloon in the rectum. The inflation is either pressure-controlled or volume-controlled. 

Regarding rectal sensitivity, the subject reports sensory perceptions such as first continuous sensation, first desire to defecate, urgency, discomfort or pain at different pressure or volume levels. The other component of rectal function evaluation, rectal compliance, is a pressure–volume relationship that refers to the rectum’s ability to stretch and expand as it receives and holds fecal matter. It contains a passive component, dependent on the mechanical properties of the rectal wall, as well as an active component via relaxation of smooth muscle in response to an increase in rectal pressure. Higher compliance means that the rectum can accommodate more fecal matter without a significant increase in pressure. While it is theoretically possible to assess rectal compliance with an elastic balloon by adjusting for its intrinsic properties, it should preferably be measured by using a non-elastic balloon with infinite compliance [5].

Rectal sensitivity and compliance help to understand the rectum’s response to stimuli and can aid in diagnosing conditions characterized by rectal hypersensitivity, such as irritable bowel syndrome (IBS), or rectal hyposensitivity, characterized primarily by symptoms of constipation and fecal incontinence [6,7]. 

The main methods used to assess anorectal function are anorectal manometry and the rectal barostat. Anorectal manometry is a widely used technique for detecting abnormalities in sphincter function and rectoanal coordination. A series of pressure measurements are performed to assess anal resting pressure, contractile function, coughing reflex and rectoanal inhibitory reflex (RAIR), as well as rectoanal coordination during simulated defecation. A standardization regarding equipment, measurement protocol and interpretation has recently been proposed by the international anorectal physiology working group (IAPWG) [8,9,10]. This protocol includes rectal sensitivity testing, which is performed by using a volume-controlled elastic balloon attached to a standardized high-resolution anorectal manometry (HRAM) catheter. There are no data available regarding the possibility of compliance testing using this method.

The gold standard for measurements of rectal sensitivity and compliance is the rectal barostat, according to the procedure described by the European COST (Cooperation in Science and Technology) action GENIEUR (GENes in Irritable Bowel Syndrome Research Network EURope) group, using a non-elastic inflatable bag connected to a pressure sensor and pump [11,12]. 

In summary, there are two different standard methods that measure rectal sensory function; however, the level of agreement between these methods is unknown. Furthermore, it is not known if rectal compliance can be assessed during rectal sensibility testing using HRAM. For this reason, we conducted a method comparison study by performing HRAM and barostat investigations on a group of healthy individuals. We hypothesize that rectal compliance can be measured during HRAM investigations, and we examined the correlations and level of agreement between the two methods. 

## 2. Materials and Methods

### 2.1. Subjects 

We recruited 26 healthy volunteers with the help of the Clinical Research Support Facility. The exclusion criteria comprised gastrointestinal disease, functional gastrointestinal disorders, psychiatric disease, anal or pelvic surgery, inclusive interventions during delivery, diabetes, concurrent or recent treatment with drugs affecting intestinal function or mood (antidepressants), nutritional supplements or herb products affecting intestinal function (probiotics), abuse of alcohol or drugs and a recent (<2 weeks) history of systemic steroid therapy or antibiotics.

All patients received verbal and written information and signed an informed consent form before any study-related procedures were carried out.

This study was approved by the Swedish Ethical Review Authority (Dnr 2021-00010, 10 February 2021, 2021-00010, 10 February 2021).

### 2.2. Study Design

Method comparison study. 

All participants presented at the motility laboratory at the hospital after an overnight fast. After bowel preparation with a tap water enema (750 mL) and digital rectal examination (during which understanding of the commands “squeeze” and “push” was confirmed), the participants were first investigated with HRAM according to the London consensus protocol and then with a standardized barostat protocol according to the GENIEUR study group. Symptom questionnaires were completed after the visit using the digital CRF platform “Smart-Trial”.

#### 2.2.1. Rectal Sensory Testing

The protocols for the HRAM and barostat procedures are presented in Figure 1.

The following sensory thresholds were used: first sensation (FS), first desire to defecate (FD), urgency (U), maximum tolerable volume (MTV) and, in the case of the rectal barostat, pain. Inflation was immediately terminated when the participant reported discomfort (HRAM) or pain (barostat) and the balloon was emptied. For the barostat method, we used the ascending method of limits protocol for rectal sensory testing [9,12].

In HRAM testing, the elastic balloons have a small residual volume when deflated, except directly after insertion, when the anal sphincter pressure ensures the balloon is completely empty. For this reason, rectal sensory testing was performed during the first inflation of the balloon (before RAIR). We used the results from 17 participants who underwent two consecutive rectal sensitivity tests with HRAM without removing the catheter to assess the effect of the combination of one stimulus provided by the first inflation and residual air in the balloon on the volumes required to elicit the sensory perceptions. The second sensory test was limited to U, and we compared the balloon volumes that induced U during the first and second inflations. For the HRAM investigations, we used an HRAM solid-state catheter (UniTip; UniSensor AG, Attikon, Switzerland) with an elastic balloon with a maximal capacity of 400 mL (Mui Scientific, Mississauga, ON, Canada) and a Solar GI HRM v9.1 (MMS/Laborie, Enschedé, The Netherlands) manometry system. For the barostat measurements, we used the Distender series II dual-drive barostat (G & J Electronic Inc., Toronto, ON, Canada), the Protocol Plus Deluxe Software version 10 (G & j Electronics Inc.) and a 600 mL rectal barostat bag (Mui Scientific, Mississauga, ON, Canada).

#### 2.2.2. Rectal Volume and Compliance

Since the HRAM balloon is elastic, the balloon volume is dependent on the applied inflation pressure. In the rectum, the pressure registered by the HRAM device is a combination of the internal balloon pressure and rectal pressure. 

The lack of available data makes it unclear if correction for the material properties of the elastic balloon is possible. Furthermore, the balloon properties may vary between different manufacturers and possibly even between batches and with age. 

To correct for intrinsic compliance and to assess variations in material properties, one atmospheric inflation from 0 to 400 mL was performed with twenty-four elastic balloons. In 17 cases, the balloons were used in the subsequent HRAM. Repeated consecutive inflations may affect the material properties, and it is not clear how long it takes for a balloon to regain its initial properties. To ensure that the balloon can return to its original properties after the test inflation, a minimum of 8 h passed between the atmospheric inflation and the investigation of the participant. In 16 balloons, we performed a second atmospheric inflation, directly after the first one, to assess the effect of the first inflation on the balloon’s internal compliance.

#### 2.2.3. Questionnaires

We used the standard questionnaires IBS-SSS (Irritable Bowel Severity Scoring System) and the GSRS-IBS (Gastrointestinal Symptom Rating Scale-IBS) to assess gastrointestinal symptoms and the Bristol Stool chart to assess stool consistency [13,14,15]. We used the Rome IV Diagnostic Criteria for IBS to confirm that none of the participants fulfilled the criteria for IBS.

### 2.3. Data Analysis

Raw data from the barostat and HRAM investigations were retrieved, and we used Microsoft Excel for data processing. The HRAM raw data consists of four or five pressure measurements for each ml of insufflated air. This procedure allows the assessment of intraballoon pressure for the whole volume range in increments of one ml. We used the results from the atmospheric inflation of 24 balloons and selected the lowest pressure value for each one ml volume increment and constructed a group-level pressure–volume curve for the intrinsic compliance of the elastic balloons. We subtracted these pressure values from the pressure values registered among the participants. The result is the intrarectal pressure–volume curve, including eventual variations in material properties. 

In the 17 cases where we subsequently used the balloons for measurement in the participants, we used the values for each balloon for an individual correction.

We used the mean of the differences between the highest and lowest pressures for each volume increment from 30 to 400 mL to assess the variation in balloon properties and the mean differences between first and second inflation to determine the effect of repeated inflation on intrinsic compliance.

Based on the results from atmospheric balloon inflations, it was clear that the pressure component from the intrinsic compliance of the HRAM balloon was higher than the actual rectal pressure. At the same time, repeated inflations of the same balloon showed a variation of 3–5 mmHg at the same volume. This means that at low pressures, small, unavoidable variations in intrinsic compliance lead to significant, large variations in the calculated compliance. We concluded that corrected values under 10 mmHg were more prone to these internal validity problems, and therefore we restricted the correction to avoid pressures under 10 mmHg during the whole inflation.

According to the GENIEUR protocol, RC in rectal barostat examinations is calculated at 50% of the individual maximum observed volume divided by the pressure at that point. There is no consensus on measuring compliance during HRAM, so we used the same protocol. To make the investigations as similar as possible, we used MTV as the maximum observed volume for both methods. In addition, RC was also calculated at 75% and 100% of the maximum volume observed with both methods, since the intrinsic compliance of an elastic balloon increases with increasing volume, which might reduce the effect of the variation in material properties.

### 2.4. Statistical Analysis

We used Spearman’s rank correlation coefficient to assess correlations between barostat- and HRAM-based compliance values and insufflated volumes at the described sensory cues. 

There are several published cut-offs for the magnitude of a correlation. However, these values are arbitrary and should be used judiciously. For example, a correlation coefficient of 0.65 could be interpreted as either a moderate or a strong correlation [16]. For this reason, variables that showed correlation factors above 0.6 were analyzed using the Bland–Altman method for measuring agreement [17]. 

According to the method described by Bland and Altman, we considered bias as a consistent tendency for one method to exceed the other, and we analyzed the 95% limits of agreement to assess if this bias has clinical implications. We used the one-sample *t*-test to verify if the differences between the methods are statistically significant and to obtain confidence intervals. We used the visual representation, as well as the Pearson correlation coefficient between the differences and means, to check for proportionality. 

We used IBM SPSS statistics software version 29 for the statistical analysis.

A stepwise approach checklist regarding the design, data analysis and interpretation of the results is provided in Appendix A.

## 3. Results

The demographics and clinical characteristics of the 26 participants are presented in Table 1. None of the individuals fulfilled the IBS criteria, but one of them was excluded from the compliance assessment because of an error in the pressure recordings.

### 3.1. Elastic Balloon Properties, Real Volume Assessment and Corrected Pressure

The balloons differed considerably with respect to their internal compliance. Throughout a complete inflation (0–400 mL), the mean pressure difference was 32 mmHg (SD ± 7 mmHg, *n* = 26). Regarding the balloons that were inflated two consecutive times, the mean pressure difference between the first and second inflations was 3 mmHg (SD ± 2.6 mmHg, *n* = 16).

### 3.2. Rectal Compliance Assessed with HRAM and Rectal Barostat

Descriptive statistics for compliance calculations are presented in Figure 2. 

Overall, the corrected and uncorrected HRAM-based compliance values were lower than barostat-based values. HRAM-based compliance based on uncorrected pressures showed a significant correlation with barostat-based compliance at 100% MTV. However, this correlation was weak. For 100% MTV for the group-level correction, 75% MTV and 100% MTV for the individual corrections, we observed significant correlation factors above 0.6. The results are presented in Table 2. Figure 3 shows the HRAM-based compliance at 100% MTV using the group-level correction, plotted against the barostat-based compliance at 100% MTV. 

Based on the correlation coefficients, we used the Bland–Altman method to analyze the level of agreement between barostat-based compliance and HRAM-based compliance at 100% MTV using the group level correction and at 75% and 100% MTV using the individual correction method. The Bland–Altman plots are presented in Figure 4. The one-sided *t*-test showed statistically significant differences between the two methods for all the variables. Measurements with the barostat yielded slightly higher compliance with a systemic bias (95% CI) of 5.1 mL/mmHg (3.9–6.2), 3.9 mL/mmHg (2.4–5.4) and 4.0 mL/mmHg (2.5–5.6) compared to the HRAM-based values. (*p* < 0.001 for all comparisons). The Pearson correlation test to check for proportionality was not significant for any of the variables (*p* = 0.6, 0.2 and 0.9, respectively).

### 3.3. Rectal Sensory Testing

Descriptive statistics for rectal sensory testing are presented in Figure 5. FS showed no significant correlation between the two methods (*p* = 0.46). FD, U and MTV showed significant correlations (*p* = 0.008, 0.043 and 0.03, respectively). However, there were no strong correlations (Spearman’s rho: 0.43, 0.40 and 0.57, respectively). Figure 6 shows the HRAM-based MTV plotted against the barostat-based MTV.

For comparison, the Spearman’s correlation factor for U between two consecutive inflations without removing the balloon was 0.78 (*p* < 0.001, *N* = 17). A Bland–Altman plot of the MTV for the barostat and HRAM, as well as for U with repeated sensory testing with HRAM, is presented in Figure 7.

Using the one-sided *t*-test, we found statistically significant differences between barostat-based MTV and HRAM-based MTV (mean difference: 65 mL, 95% CI: 33–97 mL, *p* < 0.001), but not for urgency at the first and second inflations (mean difference: 8 mL, 95% CI: −28 − +12 mL, *p* = 0.41). The Pearson correlation test to check for proportionality was not significant for any of the variables (*p* = 0.19 and 0.35, respectively).

## 4. Discussion

By performing a comparative study of the barostat and the HRAM methods, we observed moderate correlations and poor levels of agreement between the methods with regard to the variables included in the standard protocols. Collectively, our results showed that these methods cannot be used interchangeably for measuring rectal sensory function and compliance in clinical practice.

As Bland and Altman pointed out, the magnitude of the differences that are acceptable for the interchangeable use of the two methods is dependent on the clinical context. The sensory perceptions are subjective, and earlier studies using the rectal barostat showed that there is some variability with repeated testing in the same subject [11,17,18]. Pain showed the best reproducibility; however, HRAM-based rectal sensory testing does not use this parameter. We used the repeated sensory testing with the HRAM device as a guide for the magnitude of acceptable differences. 

### 4.1. The Impact of Material Properties

The material properties between the HRAM balloons varied more than expected. There were minimal volume differences between the real volumes and the insufflated volumes due to the intrinsic compliance of the balloons. However, this did not affect compliance calculations or correlations. For simplicity of interpretation, we showed results using insufflated volumes. The differences in pressures between the first and second inflations are small compared to inter-balloon properties, but repeated inflation during the preparation of the balloons should be avoided since repeated inflation can affect the balloon’s internal compliance.

### 4.2. Compliance Testing

Our results show that it is technically possible to assess rectal compliance during HRAM studies in a way similar to the barostat investigation if the inflation rate is kept constant by using a computer-controlled pump and the intrinsic compliance of the balloon is measured in order to correct for this internal compliance. As mentioned above, strict control over the number of inflations performed on each balloon is important. 

The compliance testing shows a strong correlation close to the maximum inflated volume but not at 50% of the inflated volume, which is used when rectal compliance is evaluated using the barostat. Hence, HRAM compliance testing cannot be used interchangeably with the barostat with the above-mentioned protocols. 

The expected difference in compliance testing at 75% or 100% MTV is about 2–5 mL/mmHg in favor of barostat-based compliance. We observed no proportional bias. Such variation might be expected from repeated testing using the same device. 

To interpret the level of agreement near the MTV and whether these methods could be used interchangeably in clinical practice, there is a need for data on normal values as well as values in pathological cases and variability with repeated testing of compliance at these distension volumes. 

### 4.3. Sensory Testing

Sensory perception testing with HRAM shows, at best, moderate correlations to their barostat-based counterparts. The HRAM-based MTV had the strongest correlation with the barostat-based MTV. This was nonetheless under 0.6 and suggests a poor level of agreement. Despite this moderate correlation, we analyzed the results in a Bland–Altman plot to illustrate the magnitude of the differences, as this makes interpretation easier, especially when compared to the Bland–Altman plot of U from two consecutive HRAM sensory tests. Both the 95% limits of agreement and 95% CI show a wide range and hence provide a clear indication that the rectal barostat and HRAM are not interchangeable with regard to normality, rectal hypersensitivity and hyposensitivity. We conclude that HRAM-based and barostat-based sensory testing cannot be used interchangeably. 

We attribute the lower correlations between sensory testing compared to compliance testing to the inherent difference of the stimulus, since the elastic rectal balloon is spherical with a maximal volume of 400 mL, while the barostat balloon can be considered a cylinder with a maximal volume of 600 mL, which means that the areas in the rectum that are stimulated are different and difficult to compare. 

### 4.4. Limitations

There are some limitations to our study. Selection bias cannot completely be avoided, and, in combination with the small number of participants, our cohort might not be representative of the whole population. However, this should not affect the comparison of the two investigation methods. 

Intraindividual differences and habituation with repeated investigations, which has been suggested to be an issue with rectal barostat studies, may affect the results. Data from a 10-year follow-up study with a new barostat investigation indicate that long-term barostat thresholds were stable in that group of subjects [19]. We used repeated HRAM investigations to study habituation in our study group and found no reason to assume that this could explain the poor level of agreement that we observed. 

Without knowledge of the precision of the methods, it is difficult to be certain that the Bland–Altman method for analyzing agreement is the optimal statistical tool to use. More modern statistical methods of method comparison, as described by Taffe et al., may provide a better comparison of agreement, bias and precision; however, this would require repeated investigations of the participants [20,21,22]. 

### 4.5. Clinical Aspects

Current consensus recommends rectal sensibility testing with simple balloon distension during HRAM as the first investigation and, in special cases, a rectal barostat [8]. Our results suggest that the use of these two methods results in incomparable results that make diagnostic interpretation and clinical decisions difficult.

### 4.6. Future Prospects

We suggest that the same type of balloon should be used in both investigations. A non-elastic balloon would be the most appropriate method, if this is technically possible with a HRAM catheter. This would allow for identical pressure-controlled protocols in both investigations. Future studies, which include repeated measurements, could then be performed to help establish a consensus on the standard method for assessing anorectal function.

## 5. Conclusions

It is technically possible to assess rectal compliance with HRAM by simply inflating the balloon in the atmosphere before the investigation. However, this only shows a strong correlation and possibly a good level of agreement with barostat testing near the maximum inflated volume and not with the currently recommended protocol. Furthermore, HRAM-based rectal sensibility testing only shows a moderate correlation with rectal barostat testing and only at the maximum inflated volume. 

HRAM and the barostat cannot be used interchangeably for either compliance or sensitivity testing. To overcome this problem, we suggest, if technically possible, the use of a non-elastic balloon with a fixed size and shape in HRAM.

## Figures and Tables

**Figure 1 diagnostics-14-00351-f001:**
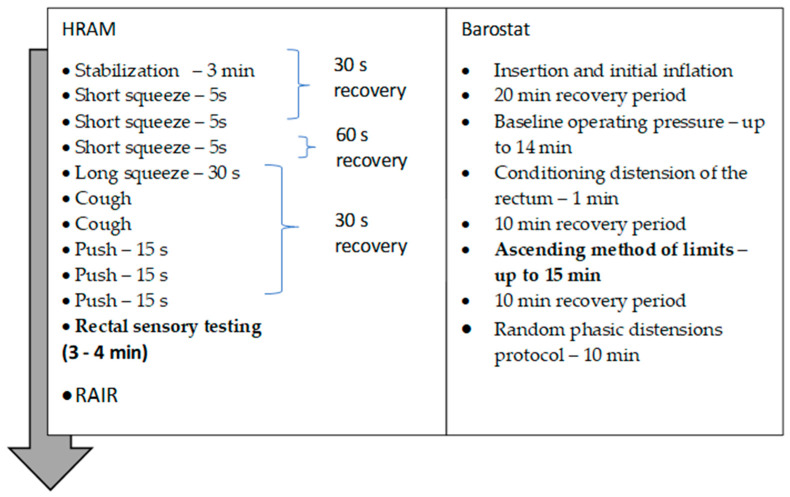
The protocols used for the HRAM and rectal barostat testing.

**Figure 2 diagnostics-14-00351-f002:**
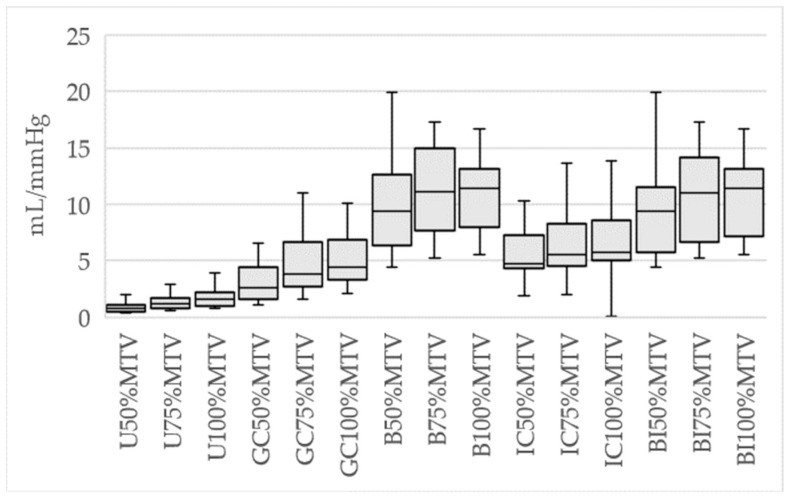
Median compliance values at 50%, 75% and 100% of the maximum tolerable volume (MTV) using uncorrected HRAM pressure (U), group-level corrected HRAM pressures (GC), individual-corrected HRAM (IC), barostat (B) and barostat investigations in the individual correction group (BI).

**Figure 3 diagnostics-14-00351-f003:**
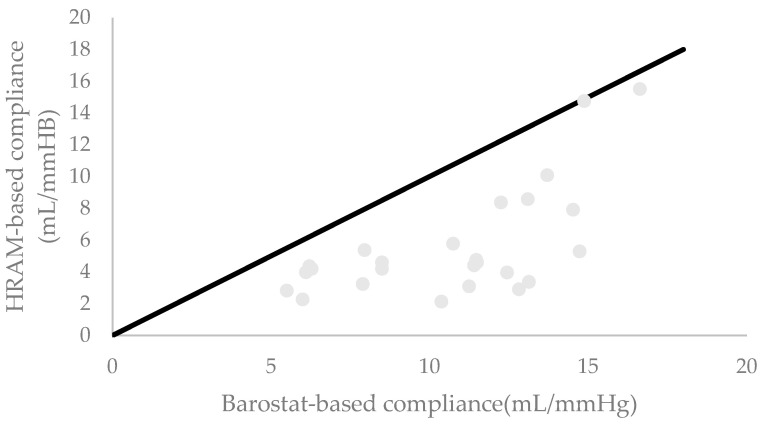
HRAM-based compliance at 100% MTV using the group-level correction plotted against the barostat-based compliance at 100% MTV. The black line represents the line of equality.

**Figure 4 diagnostics-14-00351-f004:**
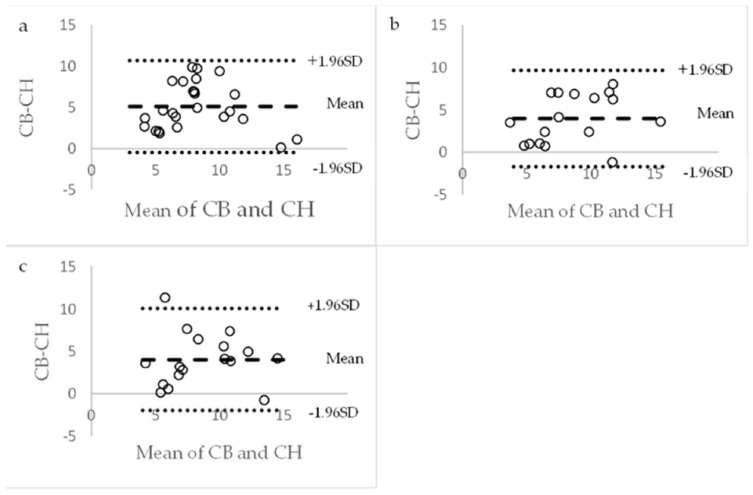
Bland–Altman plots for compliance data using (**a**) general correction at 100% MTV; (**b**) individual correction at 75% MTV and (**c**) 100% MTV. CB and CH: barostat- and HRAM-based compliance.

**Figure 5 diagnostics-14-00351-f005:**
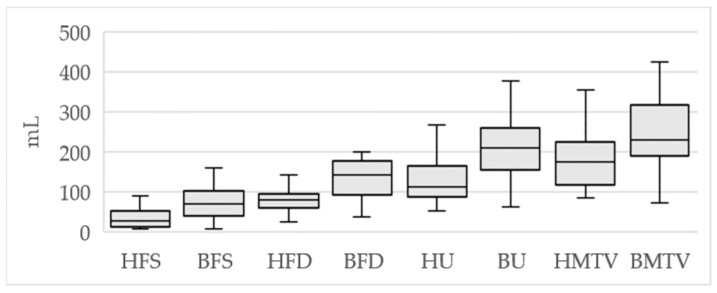
Median sensory threshold volumes with quartiles. FS: first sensation; FU: first urge; IU: intense urge; MTV: maximum tolerable volume.

**Figure 6 diagnostics-14-00351-f006:**
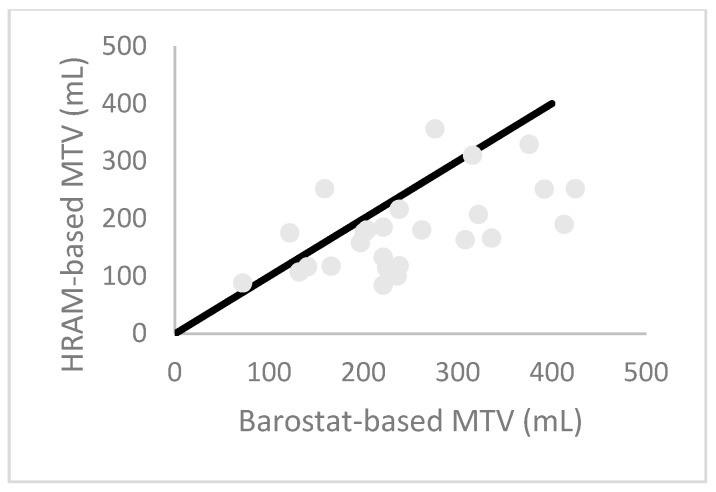
HRAM-based MTV plotted against barostat-based MTV. The black line represents the line of equality.

**Figure 7 diagnostics-14-00351-f007:**
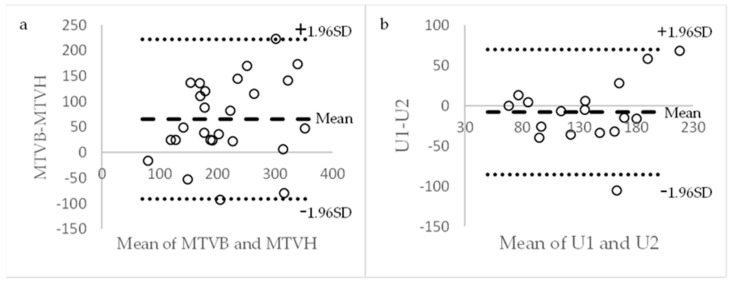
Bland–Altman Plot of HRAM-based MTV and barostat-based MTV (*N* = 25) (**a**) and HRAM-based U at first and second inflations (*N* = 17) (**b**), illustrating the difference in scale for the 95% limits of agreement. U1, U2: urgency on first and second inflations.

**Table 1 diagnostics-14-00351-t001:** Participant characteristics.

Characteristic	N (% Sample)
Male sex,	13 (50%)
	Median (IQR)
Age, years	39 (28–53)
Height, meters	1.72 (1.67–1.78)
Weight, kg	73 (66–81)
BMI, kg/m^2^	24 (22–26)
IBS-SSS, points	31 (8–58)
IBS-GSRS, points	19(14–22)
Bristol stool chart, type	4 (3–4)

**Table 2 diagnostics-14-00351-t002:** Correlation coefficients between the uncorrected and corrected HRAM pressures and the barostat values at 50%, 75% and 100% of the maximum tolerable volume (MTV).

	50% MTV		75% MTV		100% MTV	
Applied Correction	Spearman’s Rho	*p*	Spearman’s Rho	*p*	Spearman’s Rho	*p*
Uncorrected pressure	0.44	0.028	0.32	0.12	0.50	0.012
Corrected pressure	0.53	0.006	0.48	0.015	0.61	0.001
Individual correction	0.56	0.019	0.71	0.001	0.70	0.002

## Data Availability

The data presented in this study are available on request from the corresponding author. The data are not publicly available due to privacy and ethical reasons.

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
