# Peer review of "Rectal Sensory and Compliance Testing: A Method Comparison Study between High-Resolution Anorectal Manometry and Barostat Investigations"

_diagnostics, 2024, doi:10.3390/diagnostics14040351_

Round 1

Reviewer 1 Report

Comments and Suggestions for Authors

Manuscript entitled "Rectal sensory and compliance testing: a method comparison study between high-resolution anorectal manometry and barostat investigations"

This work is of interest while some modifications should be made:

1.  The authors should make the data more visualize: The authors should provide graphic data for representative cases and corelations. The highly concordent and dis-concordent cases should also be shown.

2. Table-1. The Characteristics should also include other diseases including DM, hypertension, ... etc.

Comments on the Quality of English Language

accepatble

Author Response

We would like to thank the reviewers for their time and efforts to improve our manuscript. Below we provide our response to the issues raised:

  • We added plots of HRAM measurements of compliance and rectal sensibility against correspondent barostat variables to graphically represent concordant and disconcordant cases. 
  • We recruited only healthy participants in our cohort. There are no relevant comorbidities or medications to report. 

Reviewer 2 Report

Comments and Suggestions for Authors

this is an original article concerning Rectal sensory and compliance testing.

Limitations of the study need to be added and discussed (selection bias, small number, inappropriate statistical design)

The appropriate checklist for the type of study must be added, completed and included in the methods and references.

The discussion needs to be broadened and the clinical results highlighted. A section on future prospects is necessary

The quality of the references is poor

Comments on the Quality of English Language

Minor

Author Response

We would like to thank the reviewers for their time and efforts to improve our manuscript. Below we provide our response to the issues that the reviewers raised:

  • A section on limitation regarding bias, number and study design is now added to the Discussion section.

  • The checklist for the study is now provided as a supplemental table.

  • We have broadened the Discussion and added a section on future prospects.

  • We have added some extra and more recent references.

Round 2

Reviewer 1 Report

Comments and Suggestions for Authors

The revision is acceptable for publication.

Comments on the Quality of English Language

Acceptable.

Reviewer 2 Report

Comments and Suggestions for Authors

I'm satisfied with the changes made

Comments on the Quality of English Language

Moderate